# Telomere Targeting Approaches in Cancer: Beyond Length Maintenance

**DOI:** 10.3390/ijms23073784

**Published:** 2022-03-29

**Authors:** Eleonora Vertecchi, Angela Rizzo, Erica Salvati

**Affiliations:** 1Institute of Molecular Biology and Pathology, National Research Council, Rome, Italy, c/o Department of Biology and Biotechnology, Sapienza University of Rome, Via degli Apuli 4, 00185 Rome, Italy; eleonora.vertecchi@uniroma1.it; 2Oncogenomic and Epigenetic Unit, IRCCS Regina Elena National Cancer Institute, Via Elio Chianesi 53, 00144 Rome, Italy; angela.rizzo@ifo.it

**Keywords:** telomeres, shelterins, cancer, G-quadruplex ligands

## Abstract

Telomeres are crucial structures that preserve genome stability. Their progressive erosion over numerous DNA duplications determines the senescence of cells and organisms. As telomere length homeostasis is critical for cancer development, nowadays, telomere maintenance mechanisms are established targets in cancer treatment. Besides telomere elongation, telomere dysfunction impinges on intracellular signaling pathways, in particular DNA damage signaling and repair, affecting cancer cell survival and proliferation. This review summarizes and discusses recent findings in anticancer drug development targeting different “telosome” components.

## 1. Introduction

The history of telomere was initially closely related to the concept of senescence. They were conceived as the internal biological clock limiting the proliferation potential of eukaryotic cells (Hayflick limit). It took several more years before Müller and McClintock discovered that the chromosome ends determined the Hayflick limit. 

In 1978, Elizabeth Blackburn and Carol W. Greider delved into telomeric structure, studying the protozoan *Tetrahymena thermophila*. They discovered a tandem repeated hexameric sequence that formed the telomeres of the ciliate (CCCCTT), in which the telomeric ability to protect the extremities of chromosomes resides [1]. Moreover, they proved the existence of an enzyme which is able to lengthen telomeres by adding telomeric sequences to the extremities of chromosomes. This enzyme is now known as telomerase [2]. 

In the first part of the 20th century, telomeres were discovered as the nucleoprotein structures involved in triggering senescence processes. Nowadays, the ends of chromosomes are known to play a more important role because of their implication in cancer progression. In fact, if telomere shortening is fundamental for limiting the replicative potential in normal cells, tumoral cells are characterized by telomere maintenance that determines the limitless replicative potential of cancer cells. Telomeric preservation stems from the fact that most tumor cells can activate and upregulate telomerase, blocking the telomeric shortening that would trigger the senescence or apoptosis process counteracting tumor growth. While telomere shortening can be considered a mechanism of tumor suppression because it activates senescence, it has been also related to cancer progression. In fact, in human cells, if there is an accumulation of pro-oncogenic mutations, e.g., mutations on important cell cycle checkpoint genes, cells can escape senescence, continuing to divide and increasing the likelihood of genetic errors [3,4]. Telomere fusions and chromosome breakage–fusion-bridge events unleash telomere “crises”. At this point, there are two possible outcomes: genomic instability leading to an increase of autophagy and cell death, or the crisis can be overcome by the activation of telomerase or alternative lengthening mechanisms and progression to malignant cancers (Figure 1) [5]. 

The correlation between telomeres and cancer has made it possible to characterize new molecular compounds which are able to target telomerase and telomere components for cancer therapy. Hence, in this review, we include a brief summary of old and new molecules involved in anticancer drugs targeting different telomeric components.

## 2. Telomere Evolution and Length Maintenance in Aging and Cancer

Telomeres are specialized structures at the ends of chromosome that cap and protect genetic information during cell duplication [6]. Telomeres consist of noncoding, heterochromatic, repeated DNA containing both histones and telomere-specific protein complexes [7]. Evolutionarily, telomeres are deemed to originate from intron recombinations in circular DNA molecules, generating noncoding extremities [8]. Telomeric DNA repeats are species-specific, G-C rich conserved sequences (in human 5′-TTAGGG- 3′) terminating with a G-rich (or in some species both G and C-rich) overhang [9]. 

The extremities of linear DNA molecules are not completely replicated by the DNA replication machinery; therefore, the presence of noncoding DNA at the ends of chromosome evolved to overcome the progressive loss of terminal sequences in each round of cell divisions [10]. Since telomeres are lost with cell duplication, several studies have been conducted to find correlations between telomere length and age, showing that telomere length is overall reduced with increasing age [11]. Moreover, genetic defects reducing the inherited telomere length affect offspring lifespan and the self-renewal capacity of tissues due to stem cell exhaustion [11]. 

Telomere shortening is accompanied by the presence of DNA damage response markers that individuate dysfunctional telomeres and trigger replicative senescence [12]. Mounting evidence supports a role of telomere dysfunction in human ageing-related pathologies [13]. Recently, an extensive analysis of telomere length (TL) in different human tissue types and individuals clearly showed a significant correlation of TL with genetic background, gene expression and ageing. Furthermore, telomere shortening was shown to mediate aging-related gene expression. In fact, telomeres can be shortened by exogenous mechanisms such as oxidative stress or inflammation, and a “short-telomeres” genetic signature can drive the occurrence of aging cell phenotypes [14]. 

Some cells, like gametes, cancer cells and stem cells, have developed a successful strategy to overcome the replication end problem via the expression of telomerase, a ribonucleoprotein involved in counteracting the shortening of telomeric ends. Telomerase expression is strictly controlled throughout human development; if embryo stem cells have high telomerase activity, in most adult somatic cells, telomerase is not detectable, with the exception of lymphocytes in bone marrow and peripheral blood and a cluster of epithelial cells in the skin, hair follicles, endometrium and gastrointestinal tract [15,16]. Loss of telomerase function during the embryogenesis process, generating telomere shortening right from the beginning, makes the occurrence of telomeropathies highly probable [17,18].

Telomerase is a holoenzyme which is able to maintain telomere length, resynthesizing telomeric repetitions that are lost at each replication cycle [19]. It was discovered in 1985 in the ciliate *Tetrahymena thermophila*, and was called “telomere terminal transferase” to highlight its capacity to add telomeric sequence repeats [2]. Nowadays, human telomerase structure has been defined; it is a ribonucleoenzyme formed by hTERT, the reverse transcriptase that represents the catalytic enzyme core, hTERC, the lncRNA used as a template for telomere elongation, and a series of species-specific accessory proteins, i.e., dyskerin, NHP2, NOP10, reptin/pontin, Gar1 and TCAB1 (Figure 2) [20,21]. Accessory telomerase proteins regulate telomerase activity, biogenesis and localization, and are involved in many biological processes [22]; dyskerin, for example, is a pseudouridine synthase localized mainly in the nucleus, where it can participate in the formation of telomerase, Cajal body ribonucleoparticles (scaRNPs) and H/ACA small nucleolar ribonucleoparticles (snoRNPs), playing an important role not only for telomeres, but also for rRNA processing [23].

Telomerase is reactivated in approximately the 80% of human tumors, as a mechanism of cell immortalization which is a hallmark of cancer. A small percentage of tumors (10–20%) which do not express telomerase restore telomeres length via an alternative mechanism (ALT). Preference for ALT or telomerase activation may depend on the histological origin of the tumor, the mutational background or epigenetic mechanisms. This confers different characteristics to the cancer type in terms of prognosis and response to treatments [24]. There are also a residual number of human tumors in which any detectable mechanism of telomere elongation may be found (telomere length maintenance deficient, TLM-); however, in tumor cells with ever-shorter telomeres, initial telomere length is sufficient to guarantee cell replication capacity. This demonstrates that prevention of telomere shortening is not required for oncogenesis [25].

## 3. Telomere Structure

### 3.1. Protein Complexes at Telomeres

*Shelterins*. In mammals, telomere repeats are bound by a specific complex composed of six factors: TRF1 and TRF2 (Telomere Repeat binding Factors 1 and 2) directly bind to the telomeric DNA duplex as homodimers, POT1 (Protection of Telomeres 1) binds the G-rich single-strand overhang, and TPP1, TIN2 and Rap1 act as a bridge among the shelterin factors, maintaining the structure of the complex itself [7] (Figure 2). The shelterin complex covers the telomeric DNA in a nucleosome environment and impedes the activation of repair and recombination mechanisms, allowing the cell to discriminate between natural extremities and DNA lesions. The members of the shelterin complex have distinct functions involved in different DDR signaling and repair pathways [26]; they also affect telomere elongation mechanisms [15].

When telomeres undergo massive erosion due to replicative senescence or other stresses, the shelterin complex is less abundant in the chromosomal extremities, and DDR is de-repressed, leading to cell arrest and senescence [27]. DDR at eroded and/or unprotected telomeres, failing to mask linear DNA termination, activates a signaling cascade, recruiting homologous recombination and canonical or noncanonical nonhomologous end joining machineries. These telomeres, considered as dysfunctional ones, encounter recombination events, giving rise to telomeric fusions, rearrangements or loss of telomeric repeats. 

*Nucleosomes*. The shelterin complex is not the only protein complex in telomeres. As mentioned, telomeric DNA is characterized by atypical nucleosomal organization [28]. In human cells, nucleosomes in telomeres have a shorter repeat length compared the rest of chromatin, and are less stable [29]. The telomeric nucleosomal organization seems to persist until the very end of the chromosome, limiting and affecting shelterin access to telomeric DNA [30]. Nucleosomal organization, and more generally, chromatin compaction, seem to play a role in the access of DDR factors to deprotected telomeres. Mammalian telomeric chromatin is generally considered heterochromatic, although this consideration is mainly based on observations of mouse telomeres. Indeed, the epigenetic state of human telomeres is less typically heterochromatic. More importantly, epigenetic changes of telomeric chromatin are associated with cancer progression and impact on telomeric protection, transcription, and elongation mechanisms. Trimethylation of H1 histone and HP1 binding are typical heterochromatic markers of subtelomeres and telomeres. In addition, the H3.3 histone variant, which is deposited by the ATRX/DAXX complex, is enriched in telomeres [31].

*CST complex*. The mammalian CST complex is composed of three subunits, i.e., CTC1-STN1-TEN1, and possess ssDNA-binding properties. Mammalian CST participates in the replication and maintenance of telomeres. Its depletion results in telomeric G-overhang lengthening. The complex is responsible for the termination of the extension of the telomere by telomerase late in the S/G2 cell cycle phase. In addition, it facilitates the C-strand fill-in process, which depends on the recruitment and action of pol α-primase to convert a portion of the newly synthesized telomeric end to dsDNA form. Because of its ssDNA-binding property, CST can compete with TPP1–POT1 in terms of binding the telomeric 3′ overhang and sequestering telomerase access. More recently, CST was found to be involved in replisome assembly across the entire genome, resolving stalled replication forks, and in DNA damage repair, implying that the complex is also able to bind ssDNA loops during replication [32].

### 3.2. RNA Transcription at Telomeres

*TERRA*. For a long time, telomeres have been regarded as silent chromatin territories. Recently, it was found that telomeres are transcribed into long noncoding RNAs (lncRNAs) called TERRA (TElomere Repeats containing RNA). TERRAs are transcribed from subtelomeric promoters toward the ends of chromosome; therefore, they contain sequences derived from subtelomeres followed by an array of UUAGGG repeats [33,34]. By real-time analysis, it is possible to quantify the expression of TERRAs transcribed from each specific human subtelomere [35]. TERRAs are very heterogenous in length, ranging from a few hundred nucleotides to 8–9 kilobases. The 5′ end is capped with 7-methylguanosine and the 3′ end is polyadenylated only in a small fraction (about 10%) of total TERRAs. TERRAs are transcribed by RNA-pol II from the C-rich strand, their transcription rate is modulated by the methylated state of subtelomeres, and they are strongly upregulated in cells with alternative telomere-lengthening mechanisms (ALT). They associate with telomeric chromatin both in cis- (on the same telomere from which they were transcribed) and in trans- (binding other telomeres) forming DNA:RNA hybrids. The presence of hybrids generates R-loops which increase the predisposition to hyperrecombinations of high TERRA expressing cells like ALT cells [36]. TERRAs binding to telomeres have been shown to be critical for telomere protection and stability, heterochromatin formation, telomerase regulation, homologous recombination, and length homeostasis in ALT cells (Figure 3) [37,38]. 

*telDDRNA*. DNA damage has been shown to induce bidirectional transcription starting from the site of lesion. Damage-induced long noncoding RNAs (dilncRNAs) are precursors of small noncoding RNAs (named DDRNAs) that mediate the efficient transduction of the signaling cascades driving cell arrest and repair in normal cells [39,40]. In telomeres, DNA damage or telomere shortening induce the expression of telDDRNAs that are able to mediate the expression of cell phenotypes associated with senescence [41]. To date, the contribution of DDR-associated transcription in telomeres in cancer cells has not been studied, but it may be interesting to know whether inhibition of telDDRNA could improve the efficacy of compounds targeting telomeres and inducing telomeric DDR (Figure 3).

### 3.3. Telomeric DNA and Secondary Structures

Telomere protection relies on the presence of a terminal cap-like structure called T-loop, which is stabilized by the shelterin complex (principally TRF2). The presence of t-loops at telomere ends was hypothesized almost twenty years ago based on the presence of a single-stranded overhang with sequence complementarity to the telomere duplex, and successively observed in vitro and in vivo by atomic force and super resolution microscopy [42,43]. More recently, it has been demonstrated that t-loop formation is also stimulated by telomere transcription [44]. 

Telomeres are generally difficult to replicate regions, being constituted by heterochromatin and prone to fold into secondary structures like G-quadruplexes, t-loop, I-motifs [45]. In addition, the presence of long noncoding RNA transcribed from subtelomeric promoters that stably interact with DNA duplex forming R-loops [46] enriches these chromosome fields with the topological enzymes necessary to assist replication, transcription and histone modification (Figure 4). Telomeres are indeed considered to be “difficult to repair” chromatins that consequently accumulate irreparable DNA damage causing senescence and aging [47]. In this regard, mutations affecting helicase, topoisomerase, histone acetylation and methylation cause telomere dysfunction, and consequently, aging associated phenotypes. 

## 4. Telomere Dysfunction in Cancer Initiation and Progression

In precancerous cells bearing cell cycle checkpoint failures, shortened telomere instability generates mis-segregation and chromosome breakage during mitosis, giving rise to secondary rearrangements that fuel global genetic instability [5]. Thus, telomere protection is considered to be a tumor suppressive factor. Otherwise, telomere length maintenance is a prerequisite for cancer development, since telomere attrition during cell divisions must be buffered in actively replicating cancer cells to maintain unlimited proliferative potential [48]. Telomere maintenance mechanisms are in fact considered a hallmark of cancer [49], although recently, some papers have reported the existence of human tumors without any detectable telomere elongation mechanism [25]. Moreover, a pan-cancer genomics study detected hTERT (the catalytic subunit of telomerase holoenzyme) expression in ~75% of tumor samples. In these samples, telomerase reactivation occurred by point mutation (31%) or methylation (53%) in the *hTERT* promoter [50]. Telomerase enzymatic activity is directly correlated with cancer cell proliferation and stemness [51]; reactivation mechanisms also include amplification and rearrangements of gene locus [52]. The activation of telomerase coincides with other pro-oncogene changes in adult somatic cells in the early stages of cancer development [49]. The pro-oncogenic activity of telomerase is not restricted to telomere elongation, but involves interactions between the hTERT subunit and the signaling pathways controlling cell survival and transformation like c-myc, WNT/βcatenin and NF-kB; however, the number of identified cross-talks between hTERT and intracellular signaling is constantly growing [53]. Nevertheless, current antitelomerase approaches target telomere elongation activity only, being directed toward the catalytic site of hTERT or the RNA template.

Beside telomerase and other TLM mechanisms, other telomeric proteins are found to be mutated or deregulated in cancer. POT1(Protection of Telomeres 1) is an essential component for telomere stability [54]. It binds both the ss and the dsDNA in telomeres directly, or interacting with other shelterins (namely TPP1 and TRF1) respectively; it counteracts G-quadruplex formation [55] and attenuates ATR driven DDR [56]. Germline and sporadic mutation of *POT1* are associated with different human cancers. *POT1* is frequently mutated in aggressive forms of chronic lymphocytic leukemia. Furthermore, germline *POT1* mutations have been shown to underlie a number of hereditary familial cancer syndromes involving CLL, glioma, melanoma and colorectal cancer and angiosarcoma [57]. Telomere binding proteins are overexpressed in cancer, a phenomenon which cannot simply be explained by telomere re-elongation. In fact, some aggressive cancers present an unbalance between telomere length and telomere binding protein expression, which may be the basis of the ability of dysfunctional telomeres to generate genome instability [58,59]. TRF1 is overexpressed in the early stages of pancreas tumorigenesis and glioblastoma progression in mouse models [60], and *TRF1* SNPs were found to be associated with increased risk of skin cancer in humans [61]. TRF2 is upregulated in several human cancers. It is involved in immune escape and angiogenesis through different pathways [62,63]. Alterations of the shelterin complex were recently assessed in 9125 tumor samples in 33 different human cancers. *TRF1* and *POT1* amplification and *TRF2-RAP1-TPP1* co-amplification/deletion were found to be associated with cancer progression, defining broad molecular signatures linked to several intracellular pathways involved in oncogenesis [64]. Data collected in endometrial cancer patients suggested instead an inverse correlation between TERRA expression and cancer progression [65]. 

## 5. Targeting Approaches against Telomere Components

### 5.1. Telomerase Targeting

Telomerase holoenzyme has been extensively studied as a cancer target in the last two decades by using different pharmacological approaches. One of these strategies was to compromise the catalytic activity by sequestering the human telomerase RNA (hTR) component. This strategy led to the obtainment of the only direct telomerase inhibitor, imetelstat (GRN163L), that has progressed to clinical trials. Imetelstat is a lipidated 13-mer thiophosphoramidate oligonucleotide which, by directly hybridizing to hTR with very high affinity and specificity, is able to competitively inhibit telomerase activity. When cancer cells of different histotypes were exposed to GRN163 in vitro, cellular senescence or apoptosis occurred after a period that consistently correlated with initial telomere length, needing to reach a critical level of telomere shortening. GRN163 also suppressed tumor growth in several mouse xenograft models, predominantly in a telomere length-dependent manner [66,67]. Although preclinical studies have depicted GRN163 as a promising antitumoral agent, clinical trials revealed that imetelstat failed to exhibit effective anticancer activity in solid tumors. However, imetelstat was repurposed for the treatment of myeloproliferative disorders (myelofibrosis or essential thrombocytopenia), proving effective in some hematologic patients [68,69]. Of note, in these trials, imetelstat response did not show any correlation with basal telomere length or attrition, suggesting that the mechanism of action could be due to an off-target effect rather than a telomere length-dependent mechanism [70]. According to data collected so far, on one hand, further investigations are required to determine whether short telomere length is a predictive biomarker for improving the therapeutic profile of imetelstat; on the other hand, a deeper understanding of the mechanisms behind its side-effects is needed to ameliorate the engagement of patients affected by myeloproliferative disorders.

Taking advantage of the high neo-synthesis rate of telomeres in telomerase positive cells, another anticancer strategy has been based on the incorporation of nucleoside analogues. In particular, the telomerase-dependent incorporation of 6-thio-2′-deoxyguanosine (6-thio-dG) into telomeres was shown to cause telomere dysfunction, and consequently, genomic DNA damage, cell growth inhibition and cell death of primary stem-like cells derived from brain tumors. Similar results were obtained in vivo from treatment of medulloblastoma xenografts models, thanks to the ability of 6-thio-dG to cross the blood-brain barrier and specifically target telomerase-positive tumor cells [71]. Additionally, 6-thio-dG has displayed anticancer efficacy in different solid tumors in preclinical models [71,72]. The FDA-approved anticancer agent 5-fluoro-2′-deoxyuridine (5-FdU) triphosphate, traditionally used for the treatment of liver and colon metastatic carcinomas, was recently demonstrated to be misincorporated into telomeres, rapidly inducing telomere dysfunction and cell death in telomerase-expressing cells [73]. These data support the notion that developing new, non-native nucleotide analogs to be incorporated into telomeres could represent a promising strategy to selectively target a potentially wide range of telomerase-positive cancers.

The catalytic subunit of telomerase TERT is overexpressed by cancer cells and has been considered as a potential cancer associated antigen. Indeed, endogenous TERT peptides are recognized by major histocompatibility complex (MHC) class I or II, triggering adaptive immune responses [74]. Immunotherapy approaches have been pursued in recent years targeting telomerase.

Several TERT peptide vaccines have progressed to early-stage clinical trials, showing modest cytotoxic effect; however, their efficacy remains unsatisfactory. More recently, with the discovery of immune checkpoint inhibitors that proved to have a potent antitumor immune response, improving the overall survival of patients in many cancer types, TERT vaccines have been evaluated in combination with immune checkpoint blockade in preclinical studies with promising results [75,76]. 

### 5.2. Targeting Telomeric DNA Secondary Structures

Telomeric DNA is considered to be a preferential target for **G-quadruplex** ligands, and in the last two decades, several molecules belonging to this class of compounds have shown the capacity to affect both length and structure maintenance in a dose-dependent manner [77,78]. G-quadruplexes were initially thought to act by binding and sequestering the G-overhang from telomerase elongation. In agreement with this, some G-quadruplex binders induced telomere shortening across population doublings [79,80]. Over time, new mechanisms of action have emerged that explain the short-term effects following treatment with G-quadruplex binders. Indeed, G-quadruplex stabilization can displace shelterin proteins (TRF2 and POT1) and induce a rapid DNA damage response in telomeres, triggering cell death [81]. G-quadruplex ligands also stabilize the DNA-loops forming in the telomeric duplex in the G-rich strand during replication, inducing replication-dependent damage, or transcriptional loops (R-loops) generating transcription/translation conflicts [82]. 

The synergistic effect of G-quadruplex binders with clinically employed drugs like camptothecins and PARP inhibitors, as well as ionizing radiations, has been well documented in both in vitro and in vivo preclinical models [83,84,85]. The encouraging results make this class of compounds interesting and deserving of investigation, although none of these compounds has been approved for clinical use. Notably, CX-5461, the RNA polymerase I inhibitor currently employed in clinical trials for the treatment of hematologic malignancies (Trial ID: ACTRN12613001061729), was found to bind G-quadruplex and more effectively target BRCA mutated cells [86]. On the basis of this result, currently, CX-5461 is in phase II/III clinical trial on BRCA1/2 mutated cells and BRCA1/2-deficient tumors (Canadian Cancer Trials Group ID: NCT02719997). Moreover, recent data reported the efficacy of combinations between CX5461 and PARP inhibitors, as well as topoisomerases inhibitors, as already described for other G-quadruplex binders [87]. Another very good candidate for translation into clinical practice is pyridostatin (PDS). PDS is a G-quadruplex ligand which is able to induce telomere dysfunction and also bind oncogene promoters, exerting a strong antitumor effect in vitro and in vivo [86,88]. More interestingly, PDS showed the capacity to preferentially hit BRCA1/2 mutated tumors, even in the presence of acquired PARP1inhibitor resistance, in advanced preclinical models [89,90]. The most studied G-quadruplex ligands with telomere-targeting properties are summarized in Table 1. In addition to this list of drugs, virtual-screening and mid-high throughput screening studies have revealed other classes of compounds emerging from small molecules or natural compounds libraries, with the ability to target telomeres and induce a DDR response. This confirmed that G-quadruplex interactive compounds are a continual source of new molecules with anticancer applications [91,92]. 

The C-rich strand of telomeres is known to form in vitro and in vivo quadruplex structures, namely **i-motifs**, that can coexist with G-quadruplexes or be mutually exclusive, depending on the context. Some quadruplex ligands have shown specificity for G or C-quadruplexes in in vitro binding assays, while other are selective (Table 1) [93,94]. The capacity of compounds to bind and stabilize i-motifs in vivo discriminating between G or C quadruplexes is very difficult to assess. For this reason, the ability of i-motif ligands to bind to telomeric sequences inducing DNA damage response and cell death remains elusive. Recently, an anti-i-motifs-specific antibody was developed that can help to ascertain i-motif stabilization in-vivo, facilitating the selection of specific compounds which interact with i-motifs [95]. To date, the i-motif sequence in the *myc* promoter appears to be the only biological target which is able to trigger cancer cell death [96].
ijms-23-03784-t001_Table 1Table 1Summary of compounds with in cellulo-assessed telomere-targeting activity and their mechanism of action.AgentTelomeric TargetsMechanismSynergism/Synthetic LethalityAnticancer EffectRefs.**RHPS4****and derivatives**G-quadruplexi-motifsTRF2 POT1 delocalization Replication perturbationDDR activationCamptotecinsPARPiIonizing radiationALT cellsGlioblastomaColorectal cancer[86,97]**BRACO19**G-quadruplexi-motifsT-loop disassemblyPOT1 downregulationDDR activationCis platinumLung cancerBreast cancerGlioblastoma[98,99]**Telomestatin**G-quadruplexPOT1 and TRF2 displacementG-overhang lossDDR activation ImatinibVincristinIonizing radiationGliomaNeuroblastomaSarcomaALT cellsLeukemia[100,101]**Naphthalene diimides**G-quadruplexDDR activationIonizing radiationGliomaPancreatic cancer[102,103]**Pyridostatin**G-quadruplexi-motifsDDR activationBRCA1/2 mutColon cancerRenal cancer[88,89]**Perilene coronene derivatives**G-quadruplexDDR activationN.A.Colorectal cancer[104,105]**AKT inhibitors**TRF1TRF1 downregulationN.A.Glioblastoma[106,107]**APO D41 peptides**TRF2DDR activation N.A.N.A.[108]**Curcusone C**TRF2DDR inductionN.A.Ovarian cancerEndometrial cancer[109]**Quindoline derivatives**G-quadruplexTRF2 delocalizationTERRA downregulationN.A.N.A.[110]**BPBA**G-quadruplexTERRA stabilizationN.A.ALT cells[111]

### 5.3. Targeting Shelterins

*TRF1* was found to be upregulated in glioblastoma multiforme (GBM) from human specimens and mouse models. The genetic deletion of *TRF1* impairs tumor initiation and progression in various mouse GBM subtypes by inducing telomere damage and reducing stemness in glioma stem cells, validating TRF1 as a favorable target in glioblastoma treatment [112]. Moreover, previous results showed that TRF1 inhibition impaired the growth of K-Ras-induced lung cancer in p53-deficient mice without significant side effects [113,114]. Specific PI3K and AKT chemical inhibitors were identified as being able to reduce TRF1 telomeric foci, leading to increased telomeric DNA damage and fragility, sinceTRF1 is a phosphorylation target of AKT, and these modifications regulate TRF1 protein stability and TRF1 binding to telomeric DNA [106,115]. In addition, TRF1 was found to be phosphorylated by multiple kinases involved in cell signaling pathways (ERK2, bRaf, mTOR); consequently, it is targeted by specific kinase inhibitors from an FDA approved library [107]. The repurposing of kinases inhibitors as TRF1 reducing agents has provided the rationale for proposing new combinatorial therapies based on telomere targeting in cancer. 

***TRF2*** has been implicated in several cancer-related pathways such as immune escape [116,117] and angiogenesis control through different mechanisms [62,63]. Since TRF2 is overexpressed in different human cancer types and, in some circumstances, high levels correlate with drug resistance, it has been widely proposed as a target for cancer therapy. Targeting strategies currently under development consist of peptides disrupting TRF2 protein–protein interactions. The APOD peptide was designed to mimic the TRF2 interacting domain of the exonuclease, Apollo. This peptide has been shown to induce DDR and cell death, inhibiting the TRF2-mediated recruitment of enzymes necessary for DNA metabolism [108]. On the basis of this evidence, other authors have reported the development of cyclic peptides with the same target [118,119]. 

TRF2 is known to undergo a series of post-translational modifications that regulate protein stability such as phosphorylation, SUMOylation, acetylation, deacetylation, ubiquitination, and Poly (ADP-ribose)ylation [120,121]. These mechanistical insights into TRF2 protein stability regulation provide the basis for indirect targeting of TRF2. SIRT6 deacetylases, for example, is known to stimulate TRF2 degradation. As such, its pharmacological activator could work as a TRF2 targeting agent. Extracellular signal-regulated kinases ERK1/2 regulate TRF2 phosphorylation and stability [122]; consequently, drugs interfering with ERK1/2 signaling could also play a role in TRF2 targeting. A recent drug screening revealed that at least two FDA-approved compounds (AR-A014418 and alexidine·2HCl, an inhibitor of Wnt pathway and a mitochondria-targeting agent, respectively) are able to induce antiproliferative effects by downregulating TRF2 and suppressing its pro-angiogenic and immunoescaping effects [123]. While the first compound presumably exerts its effect by acting on *TRF2* promoter (which is regulated by the Wnt pathway [124]), the second acts with unexplained mechanisms. In addition, Curcusone C, another small molecule with anticancer properties, binds to TRF2 and block its localization to DNA, inducing a DDR and cell death [109]. Finally, chemotherapeutic drugs such as arsenic trioxide (As2O3), clinically employed in the treatment of acute promyelocytic leukemia (APL), or the TopoI camptothecin, were shown to downregulate TRF2 levels [125,126].

*POT1* is the most recurrently mutated gene in the shelterin complex in cancer. Indeed, mutations affecting the interaction domains of POT1 with ssDNA or TPP1 are associated with multiple types of human malignancies such as glioma, familial melanoma, mantle cell lymphoma, chronic lymphocytic leukemia and cardiac angiosarcoma [57]. In addition, it is frequently upregulated in therapeutic and radiation-resistant cell lines. This makes it an attractive target for therapeutic intervention against POT1-related cancers. The first attempt to identify specific POT1 inhibitors came from a high-throughput time-resolved fluorescence resonance energy transfer (TR-FRET) screen for agents hampering POT1/ssDNA interaction. The yielded compound, bis-azo dye Congo red (CR), was shown to be able to competitively inhibit POT1 binding to telomeric DNA in vitro [127]. Recently, a virtual high-throughput screening (vHTS) designed against a ZINC library led to the identification of two selected natural compounds as promising inhibitors of POT1 which deserve to be further exploited as leads for the development potent and selective molecules against POT1 [128].

The possibility of adopting miRNA strategies to downregulate shelterins achieving telomeric dysfunction, and consequently cancer growth arrest, has received attention in recent years. Specifically, mir-155 targets TRF1, inducing telomere fragility, which is a specific phenotype associated with TRF1 depletion in breast cancer [129]. TRF2 and POT1 are targeted by is miR-23a and miR-185, respectively, resulting in telomeric dysfunction and cell senescence [130,131].

### 5.4. Targeting TERRAs

TERRAs play an established role in telomere protection and genome integrity (recently revised in [37]). Growing evidence not surprisingly attributes an anticancer role to TERRA [64,65,132], TERRA transcription was shown to be affected by acridine [133] or quindoline derivatives [110]. Specifically, the acridine derivative 2c induces the upregulation of TERRA levels by blocking the localization of TRF1 to telomeric sequences. Accumulated TERRA can, in turn, interact with TRF1, further destabilizing its binding to telomeric DNA. Otherwise, the acridine dimer DI26 results in the downregulation of TERRA levels by perturbing telomeric i-motifs. Both compounds exhibit antitumor activity, albeit triggered by differently altered levels of TERRA. Interestingly, TERRAs have been shown to form G-quadruplex structures in vitro that can be stabilized by cationic ions or G-quadruplex interacting compounds [132,134]. TERRA G-quadruplex are characterized by parallel structures that interact with ligands, forming bimolecular complexes different from DNA G-quadruplex complexes. Furthermore, RNA:DNA hybrids formed by TERRAs give rise to DNA loops forming G-quadruplexes on the opposite strand, increasing the complexity of the DNA topology at telomeric sites and contributing to telomere stability [135]. The structural diversity between telomeric DNA and RNA G-quadruplex presents an interesting opportunity for the development of RNA-specific interacting compounds [136]. The quindoline derivative CK1-14, instead, stabilizes TERRA G-quadruplex structures and thus induces telomeric DNA-damage response and cell death in U2OS cancer cells by interfering with TRF2 binding to chromosome ends. This compound could be considered as an inhibitor of TRF2 with promising anticancer properties in ALT tumors which are notoriously characterized by abundant levels of TERRA. Other G-quadruplex ligands have been also identified by virtual screening as TERRA binders, stabilizing telomeric RNA molecules as a consequence of sequestering them from degradation upon direct binding [111]. 

## 6. Conclusions

Telomeres are nucleoprotein complexes involved in genome stability, cell proliferation, cancer predisposition and aging. Their homeostasis is subjected to a complex interaction network comprising multiple regulation levels. The activation of persistent mechanisms of telomere maintenance which typically characterize cancer cells led scientists to develop different pharmacological approaches regarding telomerase in recent decades. Among them, imetelstat has only progressed into clinical practice for the treatment of myelodysplastic syndromes, but failed in clinical trials involving solid tumors with telomere dysfunction. DNA damage response activated in telomeres can trigger genetic instability and cell death in tumor cells, giving the rationale for further investigations into telomere targeting in cancer. Telomere-directed approaches have already been shown to preferentially kill cancer cells compared to normal ones, and therefore represent a potential Achille’s heel for cancer cells [137]. Telomeric dysfunction can be achieved through different approaches targeting DNA, RNA and proteins. Aberrant expression of telomeric components and their association with pro-oncogenic pathways and poor outcomes in cancer patients highlight new relevant targets among “telosome” components, which are worthy of further study for the development of innovative anticancer strategies (Figure 4). Recent advances in the identification of promising antitelomere strategies have been achieved by using high-throughput virtual screening, structure-based drug design and drug repurposing approaches, leading to the synthesis of new, powerful compounds to be employed as single agents and in rationale-based combinations with standard therapeutics. 

## Figures and Tables

**Figure 1 ijms-23-03784-f001:**
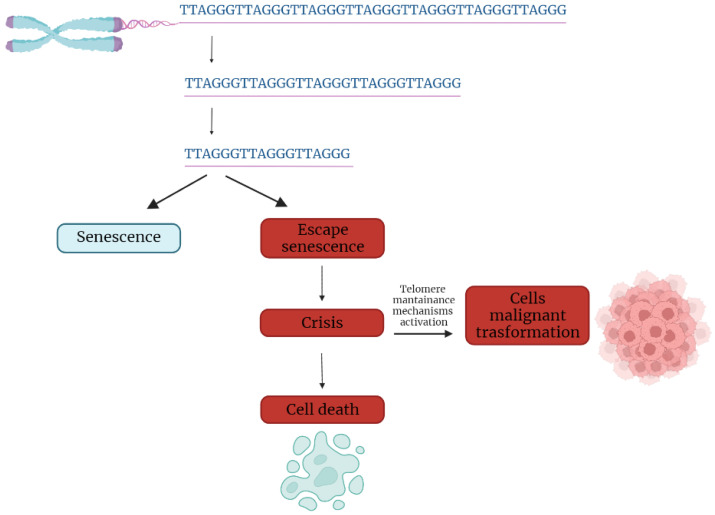
Telomere shortening triggers senescence. Escape from senescence, driven by checkpoint alteration, induces a state of “crisis” where telomeric and genetic instability lead to massive cell death. Surviving cells acquire malignant phenotypes.

**Figure 2 ijms-23-03784-f002:**
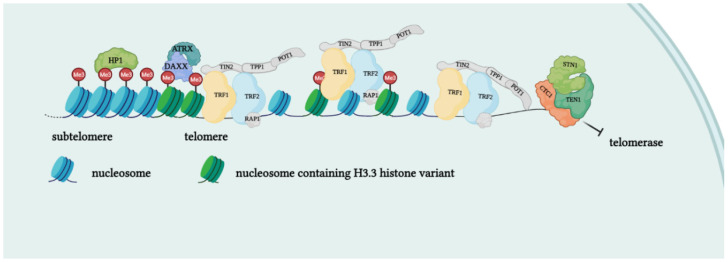
Protein complexes at telomeres. The shelterin complex binds telomeric sequences in an atypical nucleosome environment. Heterochromatic marks (histone trimethylation, HP1) and telomere-enriched histone variants (H3.3) are present. Telomeric ends are bound by the CST complex which antagonizes telomerase access.

**Figure 3 ijms-23-03784-f003:**
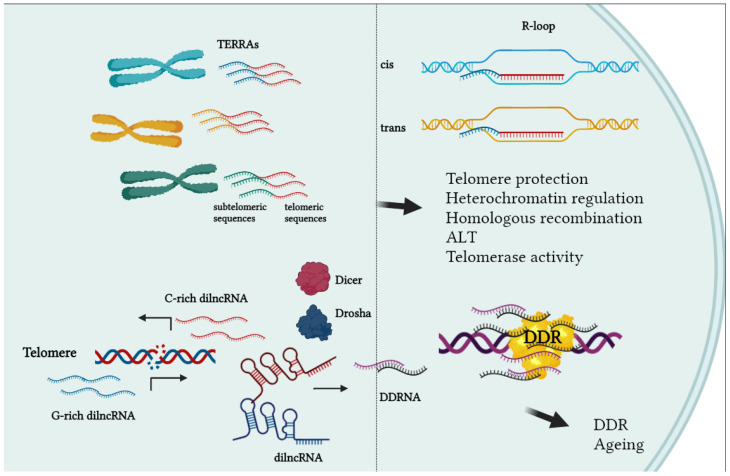
RNA transcription at telomeres. Telomeres are transcribed starting from subtelomeric regions toward telomere ends. They form DNA:RNA hybrids in cis or trans. TERRAs are involved in telomere protection, elongation and recombination. Telomere transcription is driven by DNA damage and leads to the formation of telomeric RNAs which boost DNA damage response. They are also transcribed by eroded telomeres during replicative senescence, which contain DDR markers and are involved in ageing.

**Figure 4 ijms-23-03784-f004:**
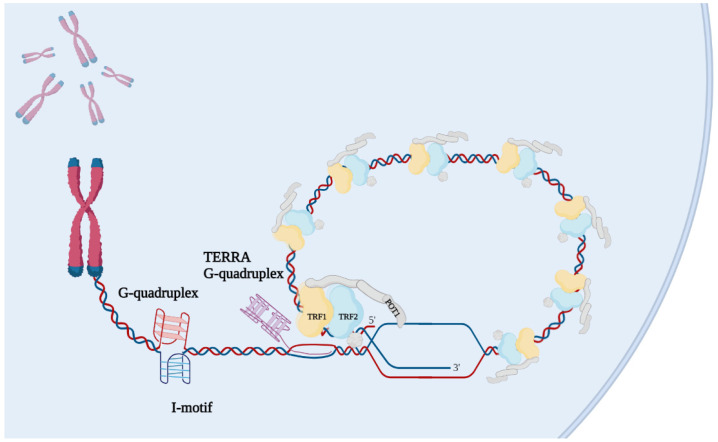
Recognized telomeric pharmacological targets. Besides telomerase and elongation mechanisms in general, the structural components of telomeres are now established pharmacological targets for the development of anticancer compounds, specifically, DNA and RNA secondary structures such as telomeric DNA G-quadruplexes and I-motifs, and TERRA G-quadruplex, as well as shelterin complex members with recognized roles in cancer progression.

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
