# Peer review of "Telomere Targeting Approaches in Cancer: Beyond Length Maintenance"

_ijms, 2022, doi:10.3390/ijms23073784_

Round 1

Reviewer 1 Report

Such a review the manuscript is too preliminary. For example, the revised manuscript does not improve clarity. My suggestion < No detailed description on TERRAs structure> is not improved. DNA and RNA G-quadruplexes activities in basic genetic processes are an active area of research in telomere. A large number of papers have been published, but none of TERRA G-quadruplex paper has been cited.

Author Response

Response to Reviewer 1 Comments

Point1 Such a review the manuscript is too preliminary. For example, the revised manuscript does not improve clarity. My suggestion < No detailed description on TERRAs structure> is not improved. DNA and RNA G-quadruplexes activities in basic genetic processes are an active area of research in telomere. A large number of papers have been published, but none of TERRA G-quadruplex paper has been cited.

Response1 We thank the reviewer for his suggestion, in the present version we improved the description of TERRAs G-quadruplex (raws 440-443) and we added more references (136-140).

Reviewer 2 Report

‍The manuscript “Telomere targeting approaches in cancer: over the length maintenance.” was submitted as a review article to be published in “International Journal of Molecular Science”.

The authors propose to summarize and discuss about the recent findings in anti-cancer drug development targeting different “telosome” components.

The revised version of the manuscript was carefully prepared and the authors introduce figures or images that facilitate and encourage its reading and comprehension.

The inaccuracies along the manuscript previously pointed were corrected and the sentences clarity and paper quality were significantly improved, in special in what concerns the introduction.

However, I still think that the authors should expand the discussion of the references. The ratio between the number of pages of discussion and of bibliographic references is still approximately 1:1. Considering that the authors propose and submit this article as a review, the discussion of the presented papers should be valuable for the scientific community.

In my opinion, the paper deserve publication in “International Journal of Molecular Science” after an improvement in the bibliography discussion and presentation, for example including schemes and figures that allow an easy and rapid access to the content of the references..

Author Response

Response to Reviewer 2 Comments

The authors propose to summarize and discuss about the recent findings in anti-cancer drug development targeting different “telosome” components.

The revised version of the manuscript was carefully prepared and the authors introduce figures or images that facilitate and encourage its reading and comprehension.

The inaccuracies along in what the manuscript previously pointed were corrected and the sentences clarity and paper quality were significantly improved, in special concerns the introduction.

Point 1 However, I still think that the authors should expand the discussion of the references. The ratio between the number of pages of discussion and of bibliographic references is still approximately 1:1. Considering that the authors propose and submit this article as a review, the discussion of the presented papers should be valuable for the scientific community.

In my opinion, the paper deserve publication in “International Journal of Molecular Science” after an improvement in the bibliography discussion and presentation, for example including schemes and figures that allow an easy and rapid access to the content of the references.

Response1 We thank the reviewer for his kind comments and suggestions, in the present version we improved the clarity of the figures by adding a more detailed legend that resumes the discussed results. In the table, we added the references in the last column to resume the most significant results related to the described agents. Further comments were added in the conclusion session, moreover some references have been discussed more extensively throughout the 5th chapter.

Reviewer 3 Report

General point

The title of this review “Telomere Targeting Approaches in Cancer: Beyond the Length Maintenance” leads the reader to expect more detail on what represents a good target and the reasons why. This is unfortunately lacking in the review.

Furthermore, there is an imbalance between the section covering general telomere biology (parts 1 to 4) and the section which addresses targeting approaches (part 5). Based on the title, the latter should be the core of the review.

Specific points

Introduction:

  • Figure 1: During a telomere driven crisis, autophagy is responsible for cell death and not apoptosis. (Nassour J et al Nature 2019)
  • Figure 1: senescence does not lead to apoptosis. Change the figure.
  • Figure 1: resolution of the image has to be improved
  • Figure 1 and line 45: escape from crisis require the activation of a telomere maintenance mechanism (telomerase reactivation or in some cases alternate mechanisms).

Part 3:

  • line 131-136: add a reference
  • Nucleosomes paragraph: add more references
  • Figure 2: improve resolution and add a detailed legend
  • Figure 3: improve resolution and add a detailed legend
  • Figure 4: improve resolution and add a detailed legend
  • Line 186-188: add a reference

Part 5:

  • Line 327: quote also the articles where they showed that G4 ligands lead to telomere erosion across PDs
  • Line 356-357: add a reference
  • Page 11: the name of the agent is missing on the first lane

Author Response

Response to Reviewer 3 Comments

The title of this review “Telomere Targeting Approaches in Cancer: Beyond the Length Maintenance” leads the reader to expect more detail on what represents a good target and the reasons why. This is unfortunately lacking in the review.

Furthermore, there is an imbalance between the section covering general telomere biology (parts 1 to 4) and the section which addresses targeting approaches (part 5). Based on the title, the latter should be the core of the review.

We thank the reviewer for his/her suggestions, we expanded the conclusion session discussing more the rational for pursuing investigating on telomere-targeting anticancer strategies. We also expanded the chapter 5 discussing more the presented references. The specific points have been addressed as follows

Point1. Introduction: Figure 1:

-During a telomere driven crisis, autophagy is responsible for cell death and not apoptosis. (Nassour J et al Nature 2019).

-Senescence does not lead to apoptosis. Change the figure.

-resolution of the image has to be improved

Response 1: The Figure 1 was modified following the reviewer’s suggestion. In the relative legend, “apoptosis” has been replaced by cell death, the resolution of the image was improved.

Point2. - and line 45: escape from crisis require the activation of a telomere maintenance mechanism (telomerase reactivation or in some cases alternate mechanisms).

Response 2: The sentence in line 45 was modified following the referee’s suggestion

Point 3: line 131-136: add a reference

Response 3: A reference was added

Point 4: Nucleosomes paragraph: add more references

Response 4: three more references (32-34) were added

Point 5: Figures 2-4 improve resolution and add a detailed legend

Response 5: All the figures resolution has been improved and detailed legends were added

Point 6: Line 186-188: add a reference

Response 6: Reference number 42 was added to raw 189

Point 7: Line 327: quote also the articles where they showed that G4 ligands lead to telomere erosion across PDs

Response 7: References 89-91 were added

Point 8: Line 356-357: add a reference

Response 8: A reference was added

Point 9: Page 11: the name of the agent is missing on the first lane

Response 9: The name of the agent was in the upper raw or the table, we formatted the file to have the entire table in a single page

Reviewer 4 Report

This review by E. Salvati and col. revises the recent knowledge on telomere biology, how it impacts on genome stability and how it can be exploited as targets for cancer therapy. It is a well written, updated and balanced review which I recommend for publication. I suggest revision of the following minor points:

  • Page 1, line 22. I couldn’t understand the meaning of the word “individuating” in the sentence. I suggest replacing it.
  • Page 1, line 25. The name of the species Tetrahymena thermophila must be in italic.
  • Figures have low resolution.

Author Response

Response to Reviewer 4 Comments

Point 1: Page 1, line 22. I couldn’t understand the meaning of the word “individuating” in the sentence. I suggest replacing it.

Response 1: The sentence in page 1 line 22 has been rephrased

Point 2: Page 1, line 25. The name of the species Tetrahymena thermophila must be in italic.

Response 2: The mistake has been corrected

Point 3: Figures have low resolution

Response 3: All the figures resolution has been improved

Round 2

Reviewer 1 Report

(1) The detailed TERRA structure must be described, because telomeric RNA is a key component of telomere machinery. These structures include telomere DNA/RNA hybrid duplex, G-quadruplex, DNA/RNA hybrid G-quadruplex etc.

(2) More references from not just one lab but other labs should be cited. (J Am Chem Soc 132:9328–9334, 2013. Nucleic Acids Res 38:5569-5580, 2010. J Am Chem Soc 130:11179–11184, 2008. Proc Natl Acad Sci USA 107:14579–14584, 2010. Chem. Soc. Rev. 40, 2719–2740, 2011.)

Author Response

We thank again the reviewer for his/her suggestion. We agree with him/her about the importance of TERRA as a new target in the anti-telomere pharmacological strategies. However, the aim of the review is to summarize the "state of the art" of  telomere targeting in the field of cancer translational research. A deeper description and discussion of proposed TERRA structures is not the object of the review and it would be strange to go into the detail of TERRA structure only, leaving out a structural description of the other targets. Nevertheless, to satisfy the reviewer's requests, we added some structural details and some of the references suggested. 

Reviewer 2 Report

An improvement in the bibliography discussion and presentation, now including schemes and figures that allow an easy and rapid access to the presented bibliography content was performed.

The paper is now suitable for publication in “International Journal of Molecular Science”.

Author Response

We thank the reviewer for his/her comments

Reviewer 3 Report

The section which addresses targeting approaches remains too superficial despite the changes/additions. The section needs to be further developed, so that there is greater detail and discussion of the targeting approaches.

The part about the i-motifs should be more detailed. What evidence show that binding to i motifs leads to telomere dysfunction and cell death ? Quote the proper paper. This was not shown in ref 106. There are actually no existing ligand thus far which is specific to i motifs.

Author Response

We thank the reviewer for his/her suggestions. In the present manuscript we made the effort to resume the recent results in the field of telomere targeting in cancer, focusing on the most promising translational research studies in in-cellulo or in-vivo models. We agree with the reviewer that the i-motifs section contained some mistakes and we corrected them. However, the other sections are, to our advise, overall complete and sufficiently balanced. We understand that you could expect a more conclusive discussion, but, this is the "state of the art" in the field and there is not any further conclusion that we can make at this stage. Nevertheless, we believe that the manuscript gives a useful overview on it, that could be appreciated by the IJMS readers. 

Round 3

Reviewer 1 Report

This manuscript has been improved. It is should be noted that TERRA DNA/RNA G-quadruplex contributes to telomere protection function. Refer to this: J. Biol. Chem. 2012 , 287, 41787-41796.

Author Response

We thank the reviewer for his/her suggestion, we included the suggested reference.

This manuscript is a resubmission of an earlier submission. The following is a list of the peer review reports and author responses from that submission.

Round 1

Reviewer 1 Report

In this review, the authors describe telomere’s structure and some progress of the ligands targeting telomere. Besides the possible biological important, telomere is being explored as a potential target for cancer therapy. The concept represents the strength and main contribution of the manuscript. However, such a review the manuscript is too preliminary and of insufficient adequately supporting by the cited papers presented in its present form.

  • There are too few figures.
  • No detailed description on TERRAs structure.
  • To further clearly understand the paper for readers, future challenges in the field should be outlined, including perspectives of both chemistry and biology.
  • Some important papers did not be cited.

Reviewer 2 Report

‍The manuscript “Telomere targeting approaches in cancer: over the length maintenance.” was submitted as a review article to be published in “International Journal of Molecular Science”.

The authors propose to summarize and discuss about the recent findings in anti-cancer drug development targeting different “telosome” components.

The paper is presented in a very compact form without the use of figures or images that could be useful and facilitate reading.

A large number of inaccuracies are identified along the manuscript concerning not only the meaning and sense of the sentences but also the punctuation.

Although a large number of references were added to the review, in most cases the authors limit themselves to referring to the articles, their content not being presented or discussed. This is evident in the low number of pages of the manuscript, if we consider that this is a review article, (19 pages) nine of them being related to bibliographic references.

Along with that, several acronyms were added, their meaning not being presented. Thus, overall, the article losses a great part of its interest to the scientific community, becoming merely a bibliographic compilation easily carried out by any researcher. 

Concerning the references session of the manuscript, typing mistakes, format inaccuracies and some duplicated references were identified and should be corrected and uniformed.

Thus, in my opinion the article, in the present form, didn't deserve publication.